# Eco-Cultural Design Assessment Framework and Tool for Sustainable Housing Schemes

**Yahya Qtaishat [1,2,*], Kemi Adeyeye [1] and Stephen Emmitt [1]**

[1]   Department of Architecture and Civil Engineering, the University of Bath, Bath BA2 7AY, UK;
     ka534@bath.ac.uk (K.A.); se394@bath.ac.uk (S.E.)
[2]   Department of Architecture, Hashemite University, Zarqa P.O. Box 330127, Jordan
*   Correspondence: Yahya.mq@outlook.com

**Abstract:** Assessment tools such as BREEAM and LEED are widely used to assess physical indicators of building performance from the micro- to the mesoscale. However, the built environment represents both intangible and tangible sets of indicators that should be understood within its context. Therefore, this project proposes a prototype Eco-cultural design assessment framework and tool to enhance the process of sustainable housing development that meets the residents' socio-cultural needs whilst avoiding unwanted environmental impacts. A qualitative research design approach was adopted. The tool was developed using data derived from interviews with 81 participants from two comparative case studies of vernacular and contemporary housing in Jordan. Results showed that indicators related to wellbeing and local culture were the most discussed by participants and were associated with sustainable architecture. The tool was designed to encapsulate these findings and evaluated for its completeness and usability by 38 architects from Jordan. Results indicate that participants had positive feedback, and they deemed the tool content useful and practical for integrating Eco-cultural design indicators within architectural practice in Jordan. The research outputs are novel and significant in that they translated qualitative socio-cultural indicators into tangible design guidelines that can be effectively incorporated into existing sustainable building assessment frameworks.

**Keywords:** eco-cultural design; framework development; sustainability assessment; sustainability indicators; tool evaluation

## 1. Introduction and Background

### 1.1. Eco-Cultural Sustainability and Its Assessment

The contemporary trends of globalisation and urbanisation created many social, cultural and economic changes in developing countries, resulting in the need for quick and new construction methods to meet this demand on housing [1,2]. These critical issues establish the need for sustainable development, as well as the need for a more sustainable built environment that requires fewer resources and less energy to function. Indeed, the sustainability of the built environment has long been a focus of research, including the development of systems for assessing and rating indicators of building sustainability such as energy efficiency, water use and waste [3]. Assessment tools such as BREEAM (Building Research Establishment Environmental Assessment Method), LEED (Leadership in Energy and Environmental Design), CASBEE (Comprehensive Assessment System for Built Environment Efficiency), and SBtool (Sustainable Building tool) are widely used to assess the tangible and physical indicators of thermal and energy performance on building scale [4]. However, the built environment represents intangibly and tangibly-inherited sets of data or indicators that should be understood and experienced interdependently based on the context [5,6]. Typically, there is a lack of inclusion

of socio-cultural indicators of human living and non-tangible non-environmental indicators of sustainability in these building sustainability assessment methods [7–9]. There is an increasing need to frame environmental issues within a broader cultural and social context, both human and natural, for improving the quality of life [10]. Further, the benefits relating to life adaptability, affordability, occupant comfort and broader social benefits need to be the focus of new sustainable housing schemes in order to move to a low carbon future [11].

Many international green building assessment and rating systems have been introduced to address contextual and regional issues. However, the same metrics, weightings and benchmarks are often used but not always customised to suit the specific region or context. In LEED, for example, the developer can only acquire four additional credits for regional priorities [12,13]. The Jordanian Green Building Guide (GBG) is a non-mandatory guide and rating system specifically for Jordan. It was developed by the Jordanian Royal Scientific Society (JRSS) and Jordan National Building Council (JNBC) in 2012 [14]. The Green building guide contains parameters and credits for assessing the building's suitability to Jordan's climate, resources, legislation, policies, policies instrument, building techniques and strategies. However, the categories, parameters and credit weightings are heavily influenced by and similar to other international green building assessment methods [15], though the guide does apply regionally-specific criteria based on the climatic region of Jordan in which the project is situated (i.e., the Jordanian desert, highlands, or valley) [14]. This only applies to energy efficiency and site treatments issues. Furthermore, Jordan's GBG has the least number and weighting for social sustainability indicators among all five assessment tools summarised in Table 1.

There is a need to deliver more environmentally sustainable housing that is socially and culturally appropriate in order to transition to a low carbon future [11]. Culture with its rules matters in sustainable development because social and economic activities reflect people's behaviour, cultural values and decisions [16,17]. Sustainable solutions are, therefore, likely to be culturally based. Intangible cultural indicators need to be measured and identified to overcome weaknesses and gaps in sustainability assessment. Defining intangible indicators will ensure better consideration and integration for the design of a sustainable built environment, its functionality and its connection with cultural and social domains [18]. Buildings and communities developed without socio-cultural considerations threaten sustainability and potentially introduce risks which could lead to disruption and disturbance to ways of life and sense of place [19,20]. Designing for socio-cultural as well as environmental sustainability is, therefore, an essential approach for coping with an ever-changing globalised world.

**Table 1.** Available socio-cultural indicators in each assessment tool.

| Assessment Tool | Available Socio-Cultural Indicators | Number of Indicators | Weight of Points |
|---|---|---|---|
| BREEAM-Communities | Demographic needs and priorities. Housing provision. Delivery of services, facilities and amenities. Public realm. Safety and walkability. Access to public transport. Local vernacular. Inclusive design. Community management. | 9 | 28 |
| LEED-(Neighbourhood Design) | Housing and job proximity. Compact and mixed-use development. Access to transit. Walkable streets. Neighbourhood schools. Connection and open community. Tree-lined and shaded streetscapes. Housing type and affordability. Access to civic and public space. Access to recreation facilities. Visibility and universal design. Community outreach and involvement. | 15 | 46 |

**Table 1.** *Cont.*

| Assessment Tool | Available Socio-Cultural Indicators | Number of Indicators | Weight of Points |
|---|---|---|---|
| CASBEE-(Urban Design) | Access to amenities, facilities and parks. Transportation. Safety and security and crime prevention. Cultural and educational facilities. Access to health and commercial facilities. Quality of housing. Management of the local society. | 7 | N/A |
| SBTool (Sustainable Building Tool) | Universal access on-site and within the building. Access to direct sunlight from living areas of dwelling units. Visual privacy in principal areas of dwelling units. Access to private open space from dwelling units. Involvement of residents in project management. Compatibility of urban design with local cultural values. Provision of public open space compatible with local cultural values. Impact of the design on existing streetscapes. Use of traditional local materials and techniques. Maintenance of the heritage value of the exterior of an existing facility. Maintenance of the heritage value of the interior of an existing facility. Impact of tall structure(s) on existing view corridors. Quality of views from tall structures. Perceptual quality of site development. Aesthetic quality of facility exterior. Aesthetic quality of facility interior. Access to exterior views from interior. | 19 | 22 |
| Jordan Green Building Guide (JGBG) | Project aesthetics. Project landscaping. Social connection. Transportation. Provision of open spaces. | 5 | 11 |

*1.2. Eco-Cultural Architecture Approach and Considerations*

Abel [21] was one of the first to use the term Eco-cultural design to describe and discuss a cultural approach to sustainability based on climate, culture and context. Eco-cultural design is the physical interpretation of the culture of a region based on ecological principles that are economically viable [22]. It adapts, uses and maximises the technological performance for locally specific needs [23]. Eco-cultural architecture thus requires a hybrid mix of local culture, material and resources, combined with modern ideas and technologies adapted to local conditions that were once available in vernacular architecture.

The indicators and metrics affecting Eco-cultural design are a complex phenomenon which can also have variations, even in a single context or region [24]. People with very different attitudes and ideals respond to varied physical environments in different ways. These interactions also differ depending on the region because of changes and differences in the social, cultural, ritual, economic and physical indicators [25,26]. Therefore, an Eco-cultural approach first needs to acknowledge the range of forces and indicators acting on a particular society and context that affects the production of the built environment in that region. These indicators include various cultural, social, historical, environmental, political, economic and religious indicators.

An Eco-cultural approach also highlights the sustainability lessons of vernacular architecture. The strength of vernacular architecture comes from its ability to blend and adapt buildings into their native context [27–29]. Accordingly, an Eco-cultural logic in architecture calls for the examination and re-use of lessons from vernacular architecture as a base for socio-cultural integration. For example, the courtyard typology represents a popular type of vernacular principle which is used to address environmental issues whilst accommodating a unique eco-cultural experience. They provided a social space where the diversity of life comes together with family and society at its centre [30]. They do this by serving as a place where all types of domestic activities can be organised as needed: a place for cooking family dinners, a place for adults to socialise, a private meeting place, a safe playground for the young children under close supervision by adults [31]. It functions as an extension of interior

rooms, or it can become a spacious room in itself, enabling it to host many members of the family and society at the same time [32,33].

Valladares [34] argued that the advancement of democratic values and decentralisation of decision making in the last decades have led to the broad recognition that the successful design of housing, neighbourhoods and cities requires the engagement and participation of residents. Users and citizens can be involved at the time in defining the sustainability targets and identifying the core criteria and indicators that are going to be assessed [35]. James [36] also suggested that citizen and community initiatives provide creative and transferable solutions to seemingly intractable social and environmental challenges. Enabling the residents to identify and design measurement systems for where they live is beneficial because they will be more invested in the reliability and accuracy of the data collected [37]. Consensus-based measurement systems can serve to diffuse conflicts within a community and establish a basis for mutual understanding and improved decision-making [24,38]. Thus, the Eco-cultural approach asserts the significance of community and user involvement during different design stages. Such an approach will enable architects to address the many social and cultural needs of users [26,39].

A comprehensive literature review to define the theoretical framework for Eco-cultural design underpinned by the principles of vernacular architecture was described in Qtaishat, Adeyeye and Emmitt, [40]. Highlighted findings include:

- There is a bias in building sustainability frameworks and assessment tools toward physical building performance indicators;
- Established cultural sustainability-related indicators are limited;
- This points to a gap in neglecting the importance of socio-cultural indicators for developing context responsive, sustainable housing [40].

## 2. Eco-Cultural Design Tool

### 2.1. Development

The nature of the tool required a qualitative, descriptive approach that utilised fieldwork and interviews [41,42]. This approach was particularly useful in this study because the researcher did not fully know the critical variables to be examined. The fieldwork was conducted during a field trip to Jordan between the 1st November 2019 and the 23rd November 2019. The tool design and development was implemented in the following stages [40]:

- A critical review of recent literature on sustainability, vernacular architecture and building sustainability assessment methods to understand research gaps and map the Eco-cultural tangible and intangible design indicators;
- In addition to documentary, survey and observational data, the primary data included 81 semi-structured interviews with inhabitants of residential dwellings of various typologies from two case study areas in Jordan. The two case studies were a pilot phase of modern housing development of King Abdullah Bin Abdul Aziz in Zarqa city and vernacular housing in the old centre of As-Salt city;
- Data gathered from the case studies were subject to rounds of coding and thematic analysis. Results from the fieldwork stage were used to refine the conceptual framework and to define the tangible spatial metrics and procedural guides within the proposed Eco-cultural design tool.

The fieldwork and interviews exposed the potential benefits and importance of following the vernacular model on social and cultural sustainability. Key findings from this fieldwork are summarised as follows [40]:

- The study found that residents and users have a different perception of sustainability from that available in most sustainability assessment frameworks and indicators;
- Results showed that indicators related to wellbeing and local culture were the most discussed by participants and were associated with sustainable architecture;

- It was also found that most sustainable building assessment frameworks and methods only focus on physical environmental indicators and have neglected socio-cultural ones;
- Jordan has unique sustainability and built environment challenges due to context-related issues which affected the final list of Eco-cultural indicators;
- The study revealed additional sustainability indicators not available in the Jordanian Green Guide for sustainable dwellings design;
- Better user satisfaction and wellbeing will help to improve the sustainability of housing schemes. For example, it could prevent any unplanned or timely changes in the dwelling layout or structure that could affect its thermal comfort or energy efficiency.

The interviews also revealed that vernacular dwellings are not compatible with the modern expectations of most of the participants. However, vernacular architecture elements such as privacy, semi-private spaces and the hierarchy of spaces are significant features that have the potential to create a balance between social interaction, culture and enhanced climatic conditions.

The derived social, cultural and environmental indicators were consolidated and interpreted into spatial and design rule configurations. Social and cultural aspects of residential environments, such as the hierarchical system of movement, privacy and social interaction, have been translated into tangible physical and spatial design qualities and associated with other sustainability and building performance indicators where applicable. Figure 1 illustrates the relationship between the produced Eco-cultural indicators visually.

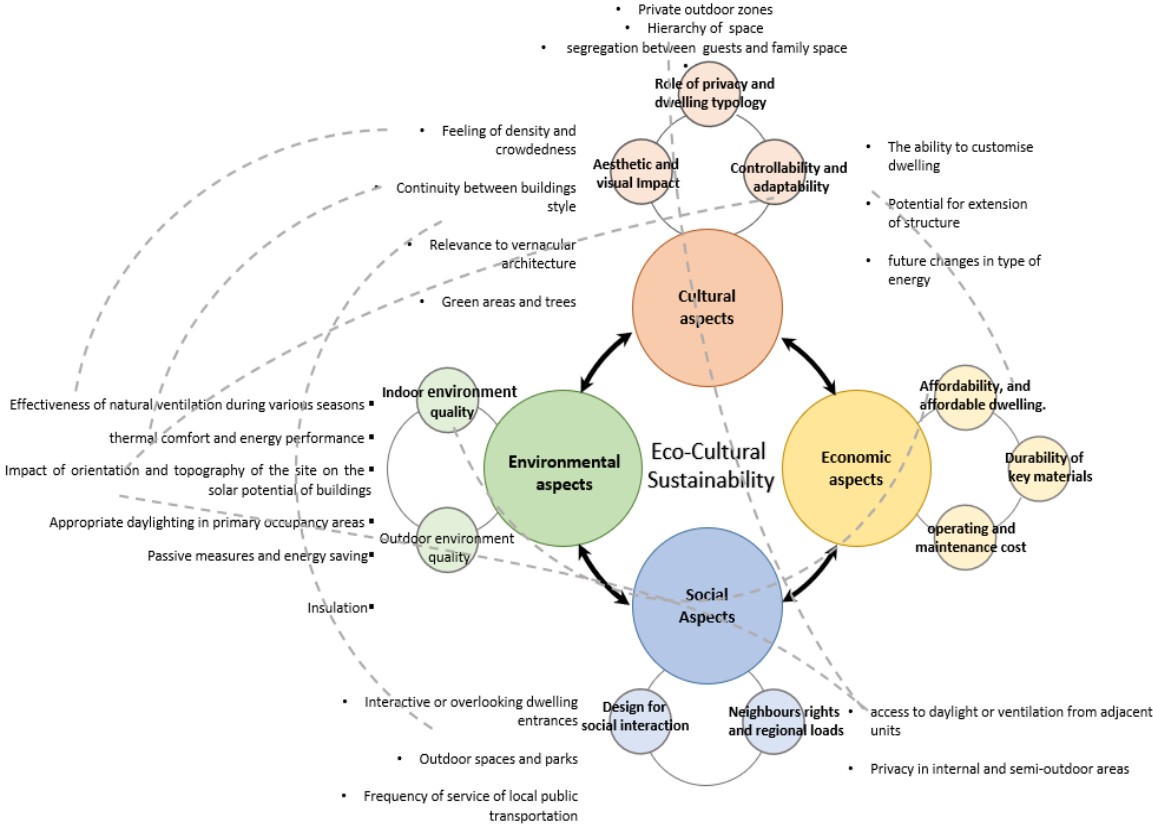

**Figure 1.** Conceptual framework of the relationships between the Eco-cultural design main indicators and categories of an Eco-cultural design [40].

This framework integrates tangible and intangible indicators, placing value on the historical, contemporary or hybrid contemporary-historic built environment. It was used to propose a new Eco-cultural assessment system for Jordanian residential buildings but can also be applied to deliver improved regional and sustainable developments in similar contexts. The framework itself is structured

along with the three aspects of sustainability and reinforces a fourth aspect: culture. The categories and indicators represent practice-related components of Eco-cultural architecture. Lines connect the indicators that directly influence one another. They also represent the relationships between themes and indicators in the context of the space and envelope of buildings. These categories and relationships were conceptualised based on the participants' viewpoints on sustainability and the quality issues concerned with new and vernacular residential dwellings.

### 2.2. Structure and Components

The Eco-cultural tool is intended to fill the critical gaps between research and sustainable architectural practice, particularly in Jordan. It is also aimed at supporting architects and urban designers to assess the environmental performance and socio-cultural suitability of proposed residential schemes, according to the resident's needs. Table 2 presents the main categories and indicators deployed in the tool. The table also highlights missing indicators in Jordan's Green Building Guide, or those that were present but were revised or adapted to reflect the findings of this study. It should also be noted that this tool represents a prototype, subject to further development.

**Table 2.** The Eco-cultural tool indicators missing from Jordan Green Building Guide.

| The Tool's Main and Sub-Categories | Missing Indicator | Adapted Indicators |
|---|:---:|:---:|
| **1. Site and context** | | |
| Flood risk | ✓ | |
| Use of vegetation to provide ambient outdoor cooling | ✓ | |
| Shading of building(s) by deciduous trees | ✓ | |
| Urban heat island effect | | ✓ |
| **2. Social Relationships** | | |
| Overlooking dwellings | ✓ | |
| Walkable streets and pathways | ✓ | |
| Proximity to services | ✓ | |
| Provision of public open space(s) | | ✓ |
| **3. Cultural and perceptual** | | |
| Visual privacy in principal areas of dwelling units | ✓ | |
| Project aesthetic | | ✓ |
| Relevance to vernacular architecture | ✓ | |
| Access to a private open space | ✓ | |
| Access to exterior views | ✓ | |
| **4. Flexibility and adaptability** | | |
| Potential for horizontal or vertical space modification | ✓ | |
| Maintenance of building components | ✓ | |
| Adaptability to add renewable energy sources | | ✓ |
| Potential for Internal space modification | ✓ | |
| **5. Indoor comfortable environment** | | |
| Effectiveness of functionality and Internal circulation | ✓ | |
| Appropriate Ventilation in primary occupancy areas | | ✓ |
| Appropriate daylighting in primary occupancy areas | | ✓ |
| Noise and Acoustics control | ✓ | |
| **6. Energy and resources efficiency** | | |
| Building orientation | | ✓ |
| Building envelope | | ✓ |
| Shading device | | ✓ |
| Use of local materials and techniques | ✓ | |
| Rainwater harvest and management | | ✓ |

**Table 2.** *Cont.*

| The Tool's Main and Sub-Categories | Missing Indicator | Adapted Indicators |
|---|---|---|
| 7. Impact on context | | |
| Solar access | ✓ | |
| Density and crowdedness (number of dwelling) | ✓ | |
| Density and crowdedness (Spaces and setbacks between buildings) | ✓ | |
| Typology and massing | ✓ | |

The prototype tool is initially implemented in Microsoft Excel with facilitated automatic scoring and point calculations based on the identified indicators and criteria. Each indicator category within the tool is further divided into sub-categories (see Table 2). Each sub-category contains sections that explain the assessment and aim to pertain it. They also cover the measures needed to achieve the objectives of the category. Other information provided includes relevant examples, links to in-tool calculators and external references to aid the users in their assessment. Figure 2 shows the initial version of the Eco-cultural tool architecture (pre-evaluation), while Figure 3 shows a screenshot of a sample page from the tool. The assessment process allows users to pre-calculate and evaluates spatial and qualitative and quantitative qualities of their design relevant to each set of indicators. Assessment is done by then choosing the quantitative value or the qualitative statement most appropriate to the design. The extent to which Eco-cultural design is achieved using the indicators are a culmination of pre-determined points based on one of four possible outcomes:

- Negative practice subtracts (−3) points from the total score (lowest);
- Minimum practice awards no points (0) to the total score;
- Good practice awards 3 points to the total score;
- Best practice awards 5 points to the total score (highest).

The proposed scale was influenced by examples presented in the SBtool framework and guide, which can be found in Laustsen and Lorenzen [43] and Cole and Larsson [44]. The indicators and sub-categories were scored using a linear scale from −3 to +5 to enable users to assess and reflect the different contextual priorities, technologies, building traditions and cultural values during the assessment process and to encourage better effort in achieving good and best practice criteria.

After completing the assessment, the last tab, entitled "Results and summary", gives a simple view of all the results, broken down by category, along with a chart. The summary aims to help address socio-cultural weaknesses in the current design proposal. Users can move back and forth between sections and categories to address issues that have low scores.

The completed assessment tool helps architects to plan a transparent and open pathway for discussing potential Eco-cultural improvements to the design with clients and the local community. This process will promote sustainable development and green building and encourage developers to build in accordance with Eco-cultural goals. The resource guide provides links to studies, research, documents, model codes/ordinances and organisations which offer additional tools and techniques for architects designing sustainable residential buildings.

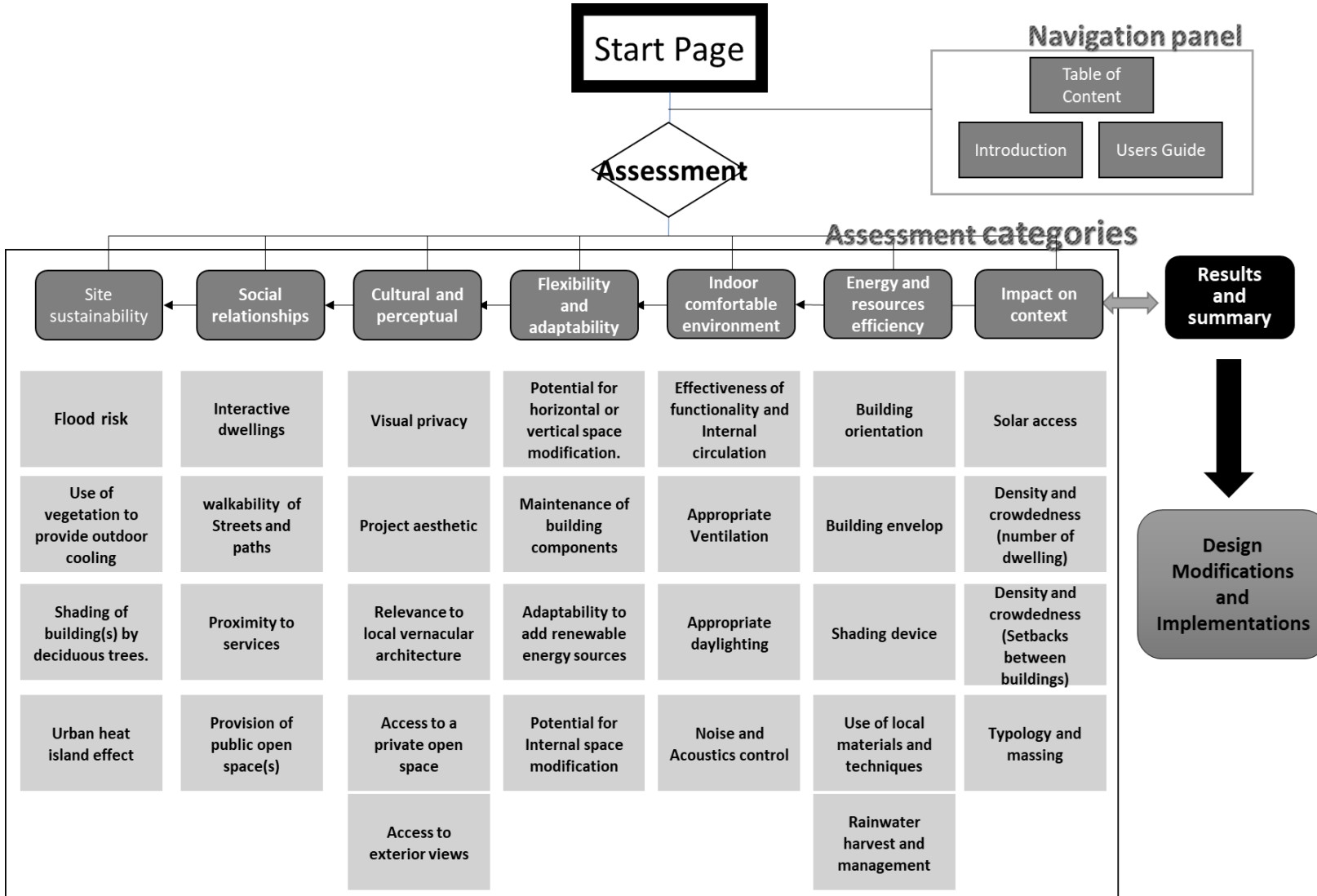

**Figure 2.** Eco-cultural tool V.1 architecture.

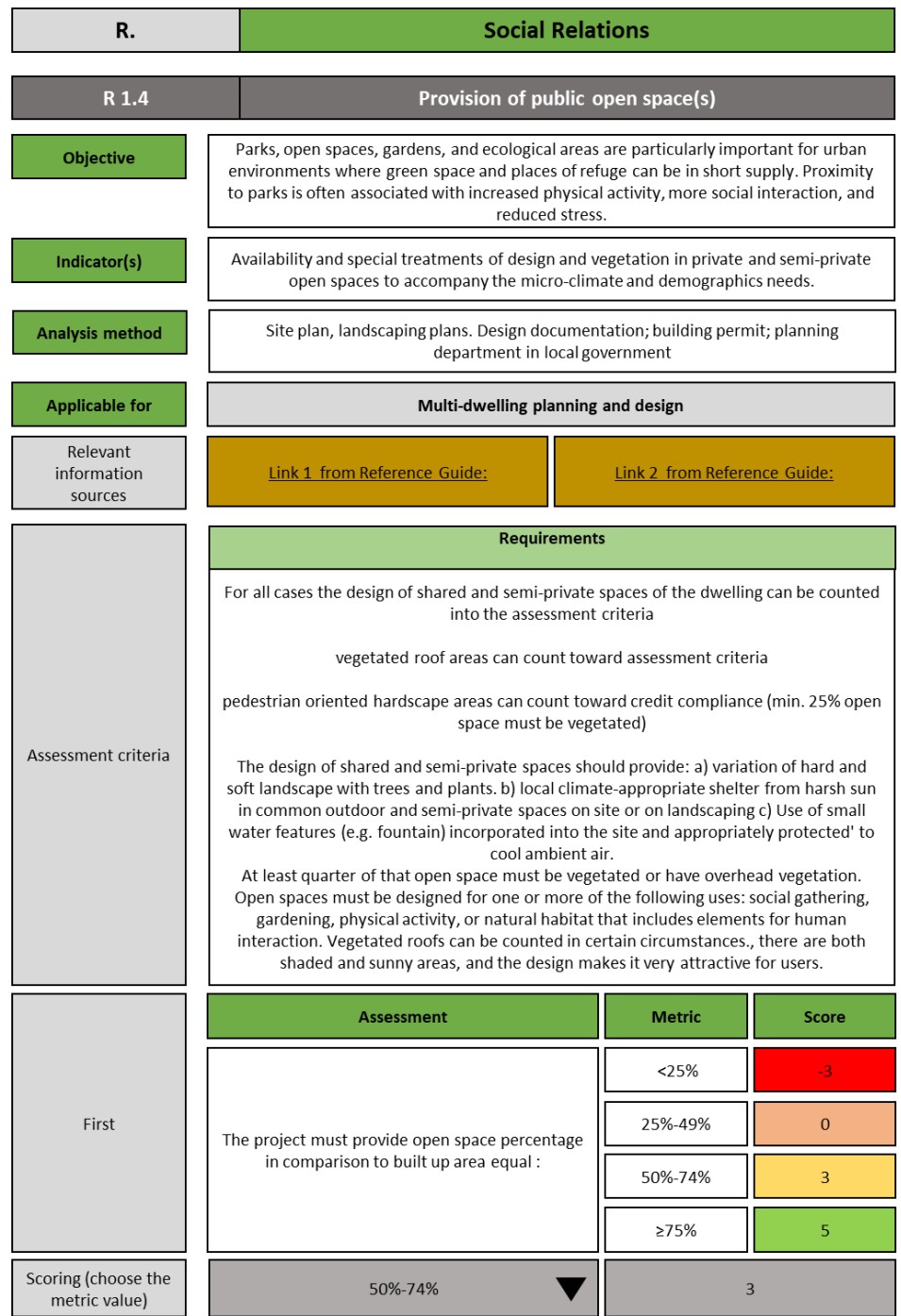

**Figure 3.** The public open space sub-category within the Social Relationship assessment section of the eco-cultural tool.

## 3. User Evaluation and Testing of the Eco-Cultural Tool

The evaluation of the compatibility between the tool and its intended use is a crucial consideration that is required in order to use an eco-design approach within a typical design workflow effectively. The tool should also be examined for use alongside standard design context, process and methods [45]. This can be achieved in collaboration with different stakeholders and users of different levels of education, knowledge and expertise [46].

An evaluation phase was needed to determine the validity of the research results in general and the potential of the tool to produce robust and usable outcomes. Evaluation with an empirical method

was used to test the tool with real users to inspect its usability and satisfaction with content [47]. The Eco-cultural tool evaluation covered three main dimensions [48]:

(a)   Effectiveness: to what extent the tool accurately presents solutions that are useful in achieving its objectives;
(b)   Efficiency: deals with the utility factors of time and effort expended to achieve the objectives;
(c)   Satisfaction: considers the acceptability of the tool's content by users.

### 3.1. Participant Sampling

Participants were recruited using purposeful convenient sampling techniques, as proposed by Patton [49] and used by Schröter [50]. The sampling was purposeful in that the potential Eco-cultural tool users were targeted by inviting participants with relevant professional experience. The sample was convenient in that emphasis was put on recruiting participants who had undertaken at least five years of professional architectural practice in Jordan, with experience of regional design, housing projects and sustainable design. The researcher started by contacting well-known architects in these fields. These architects promoted the study to other architects who could contribute to the research, adding a snowball sampling technique to the recruiting process.

Before initiating the study, a research ethics application was submitted and approved by the University. Participants were assured their identity would be kept anonymous and that they could withdraw from the study at any point or stage. All data were stored and processed based on the University's research ethics and integrity guidelines. All participants were informed about the purpose of the study, how they were expected to take part in it and how much time the experiment was expected to take. The costs to participants in the study were limited to: (a) the time involved in reading the provided script; (b) carrying out the evaluation process of the tool; (c) participating in the interview and filling in the survey.

The process started in October 2019 by the sending out of a personalised e-mail to the initial list of the pre-identified experts. Within the e-mail, the nature and purpose of the project were explained. Experts were asked if they had the time and interest to contribute to the study. In the case of the pre-selected experts, the e-mail also asked about the preferred place, date and time in order to minimise the rejection of participation in the study. After a positive response, each expert was contacted individually with a version of the final draft of the tool prepared in the Arabic language.

No specific number of participants was initially proposed. Nielsen [51] and Robinson [52] agreed that a study including at least 25 participants was quite likely to produce statistically significant findings. Faulkner [53] also suggested a minimum of 10–12 participants. Thirty-eight participants representing architects with different years and sub-fields of experience was achieved in this study, as summarised in Table 3.

**Table 3.** Summary table of research validation participants.

| Number Sampled | 38 |
| --- | --- |
| **Gender** | |
| Male | 16 |
| Female | 22 |
| **Age** | |
| 25–34 | 8 |
| 35–44 | 12 |
| 45–54 | 12 |
| 55 and more | 6 |

**Table 3.** *Cont.*

| Number Sampled | 38 |
|---|---|
| **Years of Experience** | |
| 5–9 | 4 |
| 10–14 | 8 |
| 15–19 | 6 |
| 20–24 | 10 |
| 25 and more | 10 |
| **Education Level** | |
| Bachelor's degree | 21 |
| Master's degree | 13 |
| Ph.D. | 4 |
| **Field of Expertise** | |
| Dwellings and housing design | 17 |
| Regional and vernacular architecture | 10 |
| Sustainable architecture | 7 |
| Other | 4 |

*3.2. Data Collection*

The study recruited professional architects with the intention of allowing them to try and use the tool. The aim was to assess and provide feedback on the usability and effectiveness of the content of the tool to deliver better Eco-cultural sustainable housing practices in Jordan. The purpose of this feedback was to enable the study to implement any necessary changes and add missing elements. An unstructured data collection method that included semi-structured interviews and testing sessions was used to allow for flexibility of wording and for questions to go deeper into the phenomenon. The study used Schroeter [42] and Stufflebeam [54,55] as underpinning examples by (1) incorporating an expert panel, (2) testing the tool with intended users and (3) synthesising the information from both the initial research findings and from the fieldwork. The Likert scale is a common method used in software methodology evaluation [46]. Table 4 shows the measure against each criterion. Participants assessed the statements using a five-point Likert scale, where (1) indicates 'strongly agree' and (5) indicates 'strongly disagree'.

**Table 4.** Accuracy and efficiency measurements statements.

| | Accuracy and Effectiveness of the Tool | | Efficiency and Utility of the Tool |
|---|---|---|---|
| 1. | The tool can be used effectively alongside professional practice and architectural design of dwellings. | 1. | Users need additional support to use this tool for the first time. |
| 2. | Most of the criteria and metrics are compatible with the local, context, environment and culture. | 2. | The tool is easy to learn and use in general. |
| 3. | Criteria and metrics can be implemented easily in most projects. | 3. | The tool requires a reasonable amount of time to use. |
| 4. | The tool is suitable to use at early stages of the project design. | 4. | The tool design and structure make it pleasant to navigate and use. |
| 5. | The tool has all the expected criteria to achieve Eco-culturally sensitive design. | 5. | You can be more productive using this tool. |
| 6. | Using the tool can help achieve Eco-culturally sensitive design. | 6. | Information provided with each stage was clear and effective. |

Each session was conducted individually using the same interview protocol and took approximately 60–120 min to complete. Participants were invited to use and test the tool by themselves at the beginning of the session. At the end of the test session, users completed an evaluation form, which was followed by a brief interview. The interview protocol consisted of six open-ended questions

to yield in-depth information about the tool and its potential for improvement. Reactions and comments raised by participants were also noted and documented. The six overreaching questions were:

1. What is missing from the tool?
2. What components of the tool are not necessary?
3. Are there any other errors or problems that need to be addressed?
4. What, if anything, did you like about the tool, and what did you dislike?
5. Do you have any suggestions for how to improve the tool?
6. Do you have anything else to say or add?

The questions were conceived based on the existing barriers to implementation of an eco-design approach, as highlighted in the literature review and in Qtaishat, Adeyeye and Emmitt, [40]. The session also focused on evaluating the usability of the tool alongside architectural practice. Further discussions after the testing session encouraged participants to discuss any thoughts they had while using the tool. The interviewer prompted responses with follow-up questions and explanations to gather more details about how the factors facilitated or hindered their experience and Eco-cultural perception of the local built environment. Suggestions for improvement were also requested. Written and voice notes were taken for analysis.

*3.3. Data-Analysis*

Descriptive and thematic analysis, which was backed with results from the Likert scale questionnaire to relate it to the questions of interest, and to develop rationales for improving the tool, were applied [49,56]. The thematic analysis process followed Braun and Clarke [57] and Ritchie and Spencer [58] approaches and guidelines. Data analysis was concerned with all information on (1) missing components, (2) unnecessary components, (3) errors and problems, (4) strengths and (5) means of improvement. Participants' responses also functioned as a checklist to illuminate the cultural validity of the tool. These questions were first coded for each participant. Then, responses were considered against their participants' experience and background. After that, responses were analysed by their related question group (i.e., weaknesses, errors, suggestions). Results from the questionnaire were then used to corroborate findings from participants' analysis.

## 4. Results

*4.1. User Evaluation and Ease of Use*

Figure 4 summaries the effectiveness and accuracy statements, while Figure 5 summaries the ease of use and utility. Based on participant responses and comments during the evaluation process, the results showed that most participants had positive feedback about using the tool.

Effectiveness responses scored an average of 4.17, with efficiency items scoring 3.62. The two highest-scoring statements were: "the tool is suitable for early design stages", with 4.32, and "the tool can help to achieve an Eco-cultural design", with 4.26. Both statements were related to effectiveness. On the other hand, the lowest-performing statements were "the tool is easy to learn and use", with 3.22, "the tool design makes it pleasant to use", with 3.3, and "users need additional support to use this tool for the first time", with 3.45. These results indicate that the tool performed well in content and validity, but performed less favourably in design and usability.

More than half of the participants said the tool offered architects a flexible rating system and checklist with parameters that gave them various design options and modifications according to their ideas. Moreover, "the tool is visually pleasing" statement scored 3.37, reflecting that future developments would require graphic and software design experts to follow good practice for tools of this type. The average overall score for ease of use was 3.70, which was felt to be adequate for the prototype system.

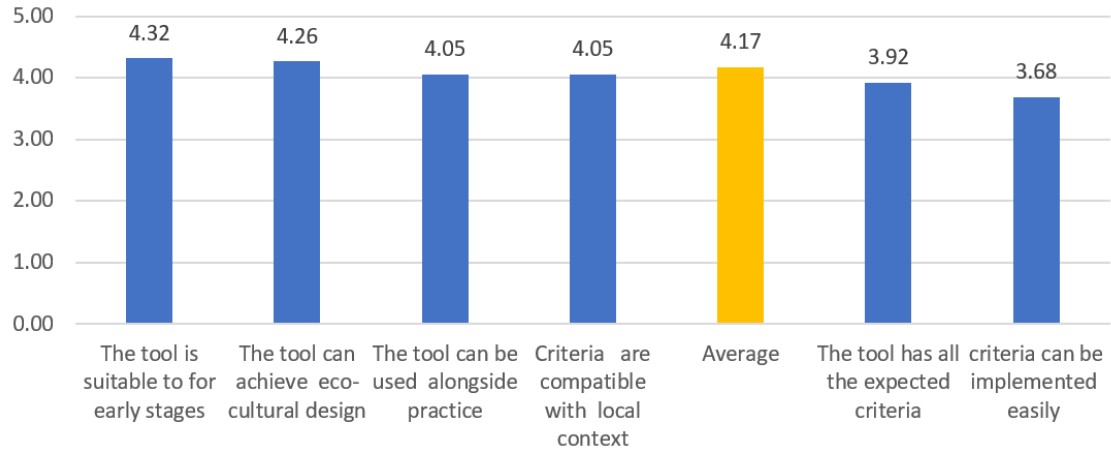

**Figure 4.** The average mean score of the effectiveness and accuracy statements.

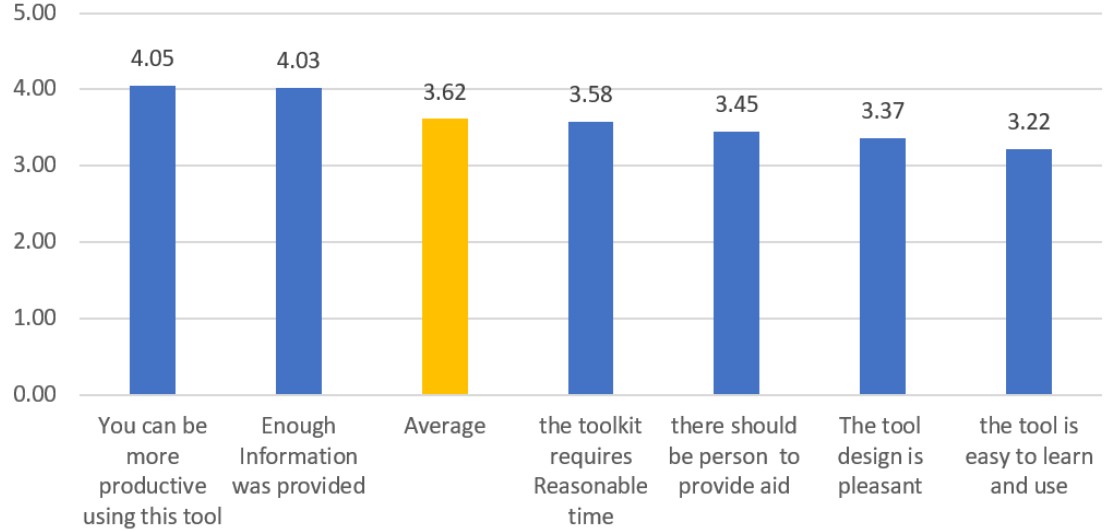

**Figure 5.** The average mean score of the efficiency and utility statements.

On the tool's ability to achieve its objectives, nearly all the participants agreed with the statement that "the tool has all the expected criteria". Furthermore, less than a quarter replied with neutral or no opinion, which indicates that the tool did not overlook essential characteristics. More than three-quarters of participants agreed to the statement "the tool can be used alongside the architectural design practice for housing in Jordan". Respondents to these questions fall within either agreed or strongly agreed, indicating that the prototype tool successfully demonstrates the potential towards a complete Eco-cultural tool.

*4.2. Content and Structure Evaluation*

The previous results were derived from interview comments and are discussed in relation to (a) characteristics of the tool, (b) potential issues of using the tool, and (c) consequences of use/problems with implementing the tool criteria and tasks. Emergent themes, issues and recommendations from the interviews are discussed in the following sections.

4.2.1. Absent Elements

As indicated in Figure 4, almost three-quarters of the participants agreed to the statement that the tool presented all the expected criteria. Figure 6 shows the "specification" or characteristics that the remaining participants considered absent within the tool.

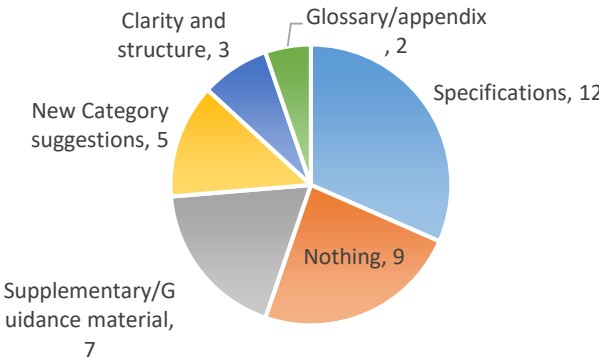

**Figure 6.** Frequency of absent themes.

Suggestions included the need for further customisation and modification. Some said that adaptability should be in the hands of the users themselves because each project would have its unique issues and parameters. However, expandability and customisation were indeed one of the objectives of the tool, which was why it was proposed in Microsoft Excel in the expectation of familiarity by most users. Furthermore, participants suggested that additional sub-tools such as checklists, templates, example matrices and indicators, footprint impact assessments and case study examples would be beneficial. Other items that occurred less frequently were the need for a "glossary" section describing the terminologies found within the tool.

4.2.2. Unwanted Elements

Almost half of the participants said that there were no parts of the tool that they felt were irrelevant to its objectives. Figure 7 illustrates the discussed themes in relation to the unnecessary aspects of the tool. The second most frequently occurring theme in the remaining responses was: "the length of the tool". A third of the participants disagreed with the statement that using the tool requires a reasonable amount of time. Participants mainly expressed their concerns that the evaluation processes and tasks might require a lengthy amount of data input and prior preparation.

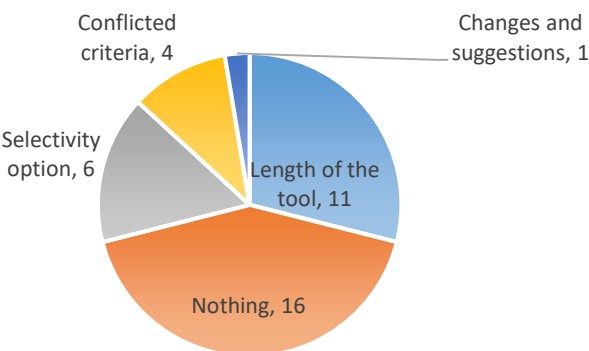

**Figure 7.** Unwanted; Frequency of Emergent Themes.

"Selectivity option" was also among the most frequently discussed ideas. Selectivity means that not all the analytical tasks might be of use for every project, and architects should be able to select which sections were relevant and ignore them if they were irrelevant. Themes such as "suggestions" and "conflict" were made less frequently. The main comments in this section revolved around a possible contradiction between some requirements and metrics in different tabs. For example, participants felt that privacy requirements could go against that of increased social interaction, or that increased setbacks in the "Solar access right" section of the tool could contradict the objective of increasing the social connection between housing units.

### 4.2.3. Errors and Confusions

Figure 8 illustrates the errors participants encountered or expressed frustration about during the trial process. Almost a quarter of the participants did not report any problems. User-friendliness was the main recurring issue, and many participants demonstrated frustration with using Microsoft Excel due to not being comfortable or confident with the software, which created difficulties. Some even tried printing it out first, but the tool was not structured for this.

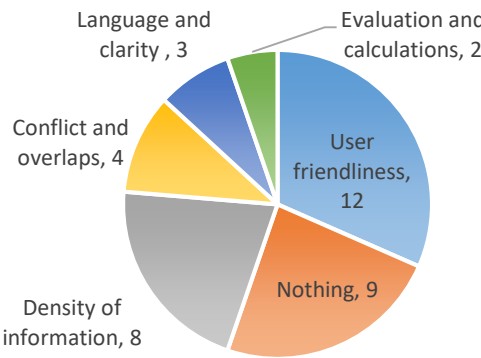

**Figure 8.** Errors and confusions; Frequency of Emergent Themes.

### 4.2.4. Strengths

As shown in Figure 9, the tool strengths are comprehensiveness and complexity, context-based approach and contribution to promote Eco-culturally sustainable housing design. Participants also associated it less with usability, raising awareness and content.

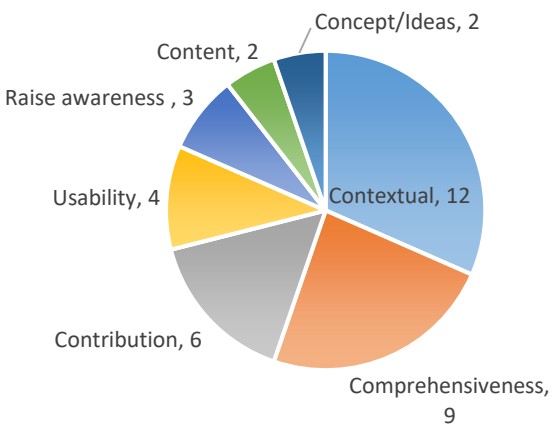

**Figure 9.** Strength; Frequency of Emergent Themes.

"Comprehensiveness" was interpreted as an indicator of the sustainably built environment covered by the tool. Participants who had experience with the green building rating systems said that the tool covered indicators that other assessment methods usually overlooked.

Almost three-quarters of participants agreed to the statement that the tool covers all the expected criteria. This statement achieved the second-best average mean score, which also supports the findings related to these criteria. Participants complemented the context-based approach of this tool. More than two-thirds agreed to the statement "the tool is suitable for context-based design in Jordan". This statement also achieved a 4.05 mean score, which was one of the highest responses among all statements.

## 5. Discussion

There has been a recent shift towards the importance of social and cultural wellbeing of occupants in research regarding a green and sustainably built environment [24,26,59–61]. However, there is a lack of direct focus in green building certification scheme guidelines to address this aspect and increase green buildings' performance in the general comfort of occupants and satisfaction dimensions [3,60]. The focus of green building sustainable building assessment tools is still geared toward empirical and quantitative issues of sustainability such as measurements of indoor air quality and thermal comfort rather than the traditional optimisation of energy performance [40,62]. The proposed strategies also lack contextualisation and adaptation to a particular region [63,64].

Results from both the fieldwork and the Eco-cultural tool use evaluation phases agree with findings of other research regarding the importance of socio-cultural indicators and context in the sustainable building assessment and design process. For example, Wen [65] reported that there is now a shift toward more inclusion of social sustainability indicators in newer versions of green building rating systems. Stender and Walter [66] and Danivska [64] argued that there is a need to evolve the sustainability assessment from mere quantitative indicators to including more abstract and qualitative indicators, such as social coherence and sense of place.

### 5.1. Content and Acceptability of the Tool

The important findings pertaining to the efficacy, effectiveness and satisfaction with the proposed tool are summarised in the following sections.

#### 5.1.1. Site and Context (Formally Site Sustainability)

The participants felt that the proposed flood treatment measures were inadequate and that architects would require more choice. Thus, more flood resilient measures were added and expanded to the site sustainability section and included in the newly added supplementary guide. Some participants also pointed out that the proposed minimum height above ground of the lowest inhabited level is less than that required in the current regulations. Therefore, the new minimum floor level was increased to the prevailing standards for projects in the flash flood zone.

The participants also felt that the tool dealt with the site and context in a two-dimensional manner. Therefore, criteria such as slope and typography were implemented in this section. Site slope is now accounted for in the metrics calculation where required within the toolkit. Finally, the participants stated that the indicators presented in the Impact on Context section overlaped with those in Site Sustainability. As a result, these sub-categories were relocated from Impact on Context to Site Sustainability. The latter was then renamed Site and Context.

#### 5.1.2. Social Relationships

New parameters were added to regulate the relationship between the outdoor-indoor zones. Participants cited increased security and visibility between the inside and outside of dwellings as essential metrics for improving social interaction. Many participants also felt the need for additional criteria aimed towards demographics to include special requirements that included children and the elderly.

#### 5.1.3. Cultural and Perceptual

The main comments in this section revolved around a possible conflict between some requirements and metrics in different tabs. For example, participants felt that privacy requirements could go against those of increased social interaction. Nevertheless, the intricate and broad nature of holistic eco-cultural sustainability requires overlapping between various sections. Therefore, similar and interconnected metrics were left without significant modification.

Finally, the participants criticised the use of old vernacular elements only. They suggested adding a page or a reference where the tool identifies factors or traits in vernacular architecture that can serve a new purpose and thus fit within contemporary design. Examples include using double facades for increased privacy and better insulation.

### 5.1.4. Indoor Comfortable Environment

Based on their experience with regional housing design, participants pointed out that there are more ways to achieve optimal circulation and privacy at the same time without following the hierarchy of spaces, such as using a zigzag or interrupted entrance viewpoint from living areas for enhanced privacy. Some participants also thought that ventilation, humidity, acoustic performance and security were missing aspects that are necessary to achieve indoor comfort. Some participants drew a connection between the objectives of this tab and environmental psychology. They suggested adding more sub-categories that handle this topic.

### 5.1.5. Flexibility and Adaptability

The lack of customisation for the tool was a key issue raised by participants. Based on many architects' experiences, they found that each project would have its unique context and issues. Therefore, the tool added customisable tabs for special site or project requirements. In addition, users can now input site and project specifications such as size, area, geographic location and climatic data in the first section of the tool. The tool presents specific requirements and parameters based on these data. This helps to provide custom tabs for that project and reduce the time and effort required to finish the assessment by removing metrics that are not required or that are incompatible with that type of project. This customisation helps to provide a more interactive and visually appealing interface.

### 5.1.6. Energy and Materials

Some of the participants felt the need to provide the users of the tool with a basic carbon footprint calculator. This can aid in visualising where the issues are in their discussions and ultimately help minimise energy and material impact. The current Jordan Green Building Guide (JoGBG) does not directly account for carbon footprint in any of its chapters, and participants felt the need for such an integrated tool. They also criticised the lack of passive heating and cooling design measures. Rating systems used in Jordan focus only on the active systems and material aspects without providing any guidance for passive solar design criteria.

### *5.2. Improvements*

Based on the findings presented in the previous section, several improvements to the tool are necessary. Table 5 summarises changes made to the tool structure and throughout its sections, while Figure 10 presents the modified tool architecture after implementing these changes. The extensive feedback provided by participants helped to resolve problems with the tool and enhance its content effectiveness and accuracy. In the longer term, the tool should be continuously improved and supplemented with clear examples, cases and training guides. More practical examples are needed to illuminate the feasibility and practicality of the tool. Additionally, the tool needs to be made interactive by developing a web-based or mobile-app format in which categories, sub-categories and indicators are interlinked and are provided with definitions to key terminologies. The tool will have to be tested and further revised according to practitioner feedback and future studies.

**Table 5.** Modifications made to the structure of the Eco-cultural tool.

| Main Findings and Issue | Actioned Modifications |
|---|---|
| Need for more Clarifications | Expanding the introduction and user guide sections. Pictures and examples have been added to the guide tab. Tabs on Excel now have colours to differentiate them from each other (blue for introductory section, yellow for assessment sections and red for summary sheet). |
| Need for a "content table" | The addition of a content table. This section also functions as a completion and progress checklist. |
| Structure | Criteria now are divided into criteria for a multi or single dwelling. Moreover, criteria are also divided into those being assessed by the architect/designer or those assessed by the owner/developer of the project. |
| Simplifications | Simplifications and adjustments to language throughout the tool. Most criteria are turned into either calculated or estimated percentages to unify the metric throughout the tool and present quantifiable rigorous metrics for assessment. |
| Terminology and glossary | Attach an appendix of all the terms used in the tool in both English and Arabic. |
| Reduce the density of the tool. | Background information and references are relocated to the end of the tool. |
| Changes to assessment criteria | Participants highlighted some needed alteration to assessment criteria to better suit the context of architectural practice in Jordan. |
| Need for more supplementary material | Adding a new section that defines and introduces various Eco-cultural strategies, architectural elements and metrics contained within the tool present an appendix with case studies and precedents of these Eco-cultural metrics. |
| Some calculation was hard for some participants | There are links to an in-tool calculator where participants can fill variables and obtain the required calculated value for some sections. |
| Overlapping | Overlaps will not be fully addressed due to the overlapping nature of Eco-cultural sustainability. Some participants felt that there is a conflict between the criteria in context and neighbour tab and between other sections. They also thought that this section as a stand-alone section was not required. So this section was removed and its sub-categories merged with other sections. |
| Fix issues of usability and ease of use | The tool needs to be made interactive by developing a web-based format in which categories, sub-categories and indicators are interlinked to make it easier and simpler to use. |

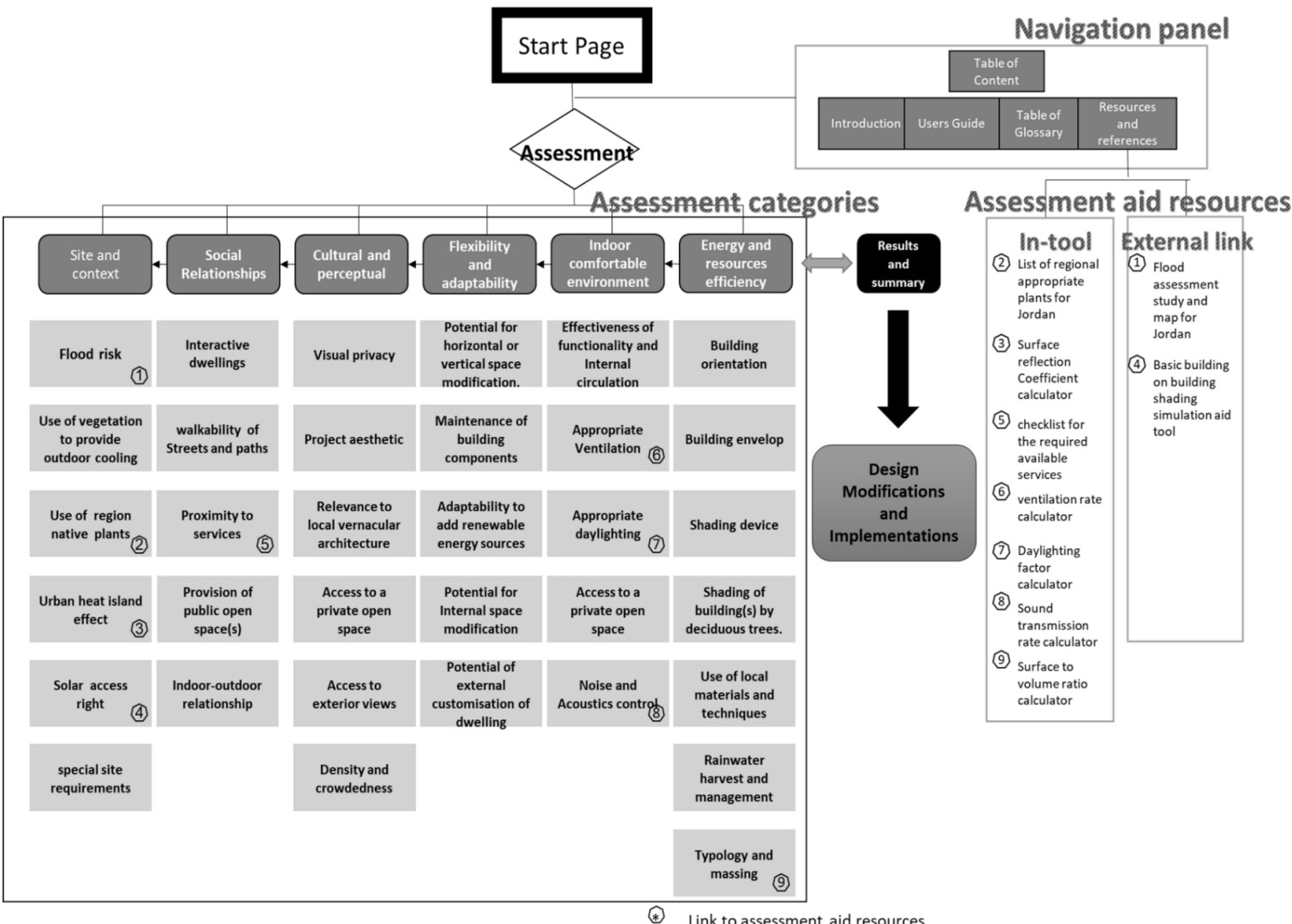

**Figure 10.** Eco-cultural toolkit V.2 architecture (Post evaluation).

## 6. Conclusions

The fieldwork results of this study found that residents prioritised design indicators related to socio-cultural appropriation and linked them to their point of view of sustainability. The fieldwork and interviews exposed potential benefits and the importance of following a vernacular model on social and cultural sustainability. A prototype Eco-cultural design assessment tool was created and here discussed to try and fill these gaps of research in an earlier stage for the contexts of Jordan.

Results of user evaluation of the tool revealed the following:

(a)    Effectiveness: Participants praised the context-based approach toward sustainability and agreed that the current assessment methods were not entirely suitable for Jordan in their current form. Participants also commended the use of socio-cultural indicators as a design moderator and as the right way for the consideration of specific residents' needs within the sustainable built environment;

(b)    Efficiency: Participants expressed that the tool allowed them to think dynamically with multidimensional constraints rather than traditional methods that focused on a single design solution;

(c)    Satisfaction: Most participants had positive feedback about the tool content. However, they were less satisfied with the usability and design of the tool.

Although the study addressed major indicators for socio-cultural sustainability in residential buildings, there are other factors that may have direct effects on the level of satisfaction depending on the region or context. Indicators such as quality of finishes and materials, quality of services and culturally related indicators such as the role of privacy should be investigated separately in other contexts and are important issues that need to be incorporated into sustainability assessment frameworks. Furthermore, the effect of applying different building codes and regulations between Jordan and other regions should also be investigated as well. However, results deducted from the case of Jordan are expected to be applicable to countries in the Middle East and North Africa region (MENA) as they share a similar climate, history, cultural and social norms [67]. It is also important to investigate other stakeholders and potential tool users in the future, including planners, engineers, potential developers and governmental officials in Jordan. This was not included within this evaluation stage as it was outside the scope of this work and also because of practical and time constraints. Further investigation is also required to test the latest version of the tool and to eliminate any unresolved usability issues. Other main research conclusions and participant comments included:

- In general terms, the participants praised the effort to make a contextual green building assessment tool for Jordan that is not based on international green building rating systems such as LEED.
- Some participants said the main barrier against implementing some of the criteria is the economic burden that the developer might face. The increase in construction and design costs would limit developers and investors from following some of the metrics.
- Participants said that the tool should provide more economic-related indicators that could relate to both developers' and residents' needs.
- Participants expressed that it would be better if the tool were more graphical, quoting that "architects are visual thinkers", or more interactive, with a web-based design and a more interactive interface.
- Participants suggested adding an index page along the tool pages to help guide the users and show their progress.
- Participants enjoyed using the provided in-tool calculators that could help measure the degree of achievement of metrics. They suggested that the absence of the equations and calculations in the tool and in similar existing tools needs to be addressed.
- There was a consensus among participants that there should be various versions of the tool that are targeted at other stakeholders such as project developers and investors.

- There were concerns by some participants that the impact of the context section had repeated ideas or indicators that overlapped with other sections. This tab was removed from the updated version, and its sub-categories were relocated to other sections.
- Participants supported the pre-chosen indicators, and all of the pre-selected indicators were preserved within the category.

This study makes a conceptual and practical contribution as it helps to bridge the gaps in knowledge by drawing relationships between intangible socio-cultural indicators and physical design guidelines to be integrated into building sustainability assessment frameworks. This research is considered to set a new precedent in research for Jordan and beyond. It deals with issues and queries intended for both academic research and practical application. Academically, this research aids in understanding the role of socio-cultural indicators in sustainable design. In practice, the toolkit was deemed useful and usable for the integration of socio-cultural indicators within architectural practices in Jordan. Additionally, the research paradigm and approach are repeatable for other contexts and regions. The tool will help its users to incorporate necessary Eco-cultural design criteria and make changes to allow for more sustainable design and green building. In doing so, local governments would find ways to encourage developers, contractors and design professionals to plan for and use sustainable design tools and techniques.

**Author Contributions:** Y.Q., contributed to the data collection and analysis of the results and to the writing of the manuscript. K.A. and S.E. contributed to the design and implementation of the research, writing and editing of the manuscript. All authors have read and agreed to the published version of the manuscript.

**Funding:** This paper is part of an ongoing Ph.D. research project at the University of Bath funded by The Hashemite University in Zarqa, Jordan.

**Conflicts of Interest:** The authors declare no conflict of interest.

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
