# Peer review of "Eco-Cultural Design Assessment Framework and Tool for Sustainable Housing Schemes"

_urbansci, doi:10.3390/urbansci4040065_

Round 1
Reviewer 1 Report
Please see the comments on the enclosed file (notes in the .pdf file)

Author Response
Point 1: These couple of sentences require citation. Literature is broad about research works assessing environmental sustainability in building sector. Reviewer suggest a couple of works that is published in MDPI series: https://www.mdpi.com/2071-1050/10/10/3469
Response 1: Thank you. Proper referencing has been added including using the works that you have suggested. (Through the introduction. Lines 35, 37,41 and lines 43-48)
Point 2: Please report here a reference
Response 2: Thank you for pointing this out. The reference for the findings is now reported before the bullet point list. (Line 143)
Point 3: How this picture has been retrieved? Is it possible that somebody else can replicate this relationship, or it was retrieved by the expertise of the authors?
Response 3: Yes, figure 1 consolidates the findings from Table 2 which has now been removed. It collates and presents the knowledge derived from the primary data within the context of what is already known and established by other researchers, and the new contributions derived from this work.
The diagram is meant to show that the work presented in this chapter is underpinned by previous theoretical and primary findings. It also shows how the proposed eco-cultural approach helps to bridge the gap between the tangible and intangible factors of the sustainable built environment. Figure 1 outlines the research design for this paper. By bringing attention to the unspoken indicators of physical space, embodied sustainable elements of the home and what people said about their homes, all of which are central to an eco-cultural approach. And equally can help other researchers duplicate this method and customise it based on their context and cultural settings. (Lines 169-178). Further discussion on duplicating and generalisation of the research results is also added ( 511-519 and 555-556)
Point 4: Information about score aggregation (mathematical equations - mean, RMS, others? - used to collect all the scores should be provided here). How this scoring matrix has been selected? Which criteria has been used ? Why only four scores? Please justify
Response 4: Thank you for your comments. We agree that the text didn't justify the scoring adequately. Justification has been added, we hope that this is now satisfactory. (Lines 208-212)
Point 5: Caption of Figure 3
Response 5: Thank you for pointing this out, we have corrected Figure 3 caption.
Point 6: For the evaluation of software performances, the reviewer also suggest to have a look at the following paper;
https://www.tandfonline.com/doi/abs/10.1080/19397038.2018.1439121
This could be helpful in the evaluation of software (including usability, effectiveness, etc.)
Response 6: Thank you for your suggestion, we have upgraded and further improved the argument for our evaluation process with this more recent literature. (lines 233-237 and Lines 282-285 and lines 300-302)
Point 7: English language and style are fine/minor spell check required
Response 7: Thank you for notifying us. the manuscript was rechecked and corrected for typos and spelling errors. (Figures 4 and 5)
Point 8: were this new "prototype" tested? If not, why to report. It would be better to discuss possible improvements in details.
Response 8: Thank you for your question. The Improvements to the tool are now discussed in detail. It was also thought it useful to show this diagram as it gives a concise representation of what has changed as a direct result of the evaluation process. This helps to further enhance the reproducibility of this work. The new prototype will be further evaluated in the next stages of the study. (lines 424 – 491)
Point 9: Authors should report in the conclusion the limitations observed conduction the analysis only in the Jordan boundaries? Is it possible to enlarge the use of the tool outside these boundaries?. If it is possible, which criteria should be stressed?
Response 9: Thank you for raising this critical issue. We have now reported the limitations of the study and clarified the generalisability of the results and methods. (Lines 511- 524)
Point 10: Please add feature works on this aim.
Response 10: the paragraph has been restructured to better justify this statement. (Lines 553-565)

Reviewer 2 Report
In the introduction, perhaps a measured discussion of the Jordan Green Guide is in order, particularly how it relates to other countries.
In your discussion on vernacular architecture (lines 70-74), a few examples of what this is might be helpful to the audience.
In the paragraph about decentralizing decision making (lines 75-80), you might want to consult some of the literature regarding charrettes.
Table 1 has some ambiguity within its design. It is not clear where the aesthetic and visual impact section ends and the role of privacy and dwelling technology begins.
Some of the typologies within this table are a little confusing as well. For example, why are the location of parking spaces part of the social aspects?
Figure 1 looks to be just a visual re-orientation of the material presented in Table 1...can you confirm that? It actually looks fine, but I am just wondering if it is a redudancy.
Also, Table 2 seems to be an expansion of Table 1; however, you seem to have left off the lower half of the categories mentioned within the earlier table.
On line 158, replace with "Table 3 presents..."
The presence of the term "modified indicators" in Table 3 requires further explanation. Was this concept mentioned in Jordan's Green Guide, or not? If so, how was it modified?
From lines 174-177, the awarding of points seems somewhat arbitrary. Please explain the rationale for the scoring heirarchy here.
Figures 2 and 10 seem to have some issues with visual quality.
When considering the criteria for the three main dimensions of the tool evaluation, it would appear that "c" would be highly correlated with "a" and "b".
On line 216, please add the word "and" before "how much time".
On line 225, please address the phrase "draft of the final draft".
On line 229, I believe that you meant to use the word "experience".
In Table 5, it would appear that item 1 has a statement that requires reverse coding compared with the other statements from the survey, based on the Likert scaling explained in the section above it...is this correct?
You have the term "discussion" in both sections 4 and 5.
While the manuscript overall is pretty well-written, there are still some issues related to spelling that can be properly addressed by going through the document a little more thoroughly. The word "average" in Tables 4 and 5 is just one example of this.
On line 389, the phrase "else of its origin country" seems awkward.
Author Response
Point 1: In the introduction, perhaps a measured discussion of the Jordan Green Guide is in order, particularly how it relates to other countries.
Response 1: Thank you for this comment and we agree that this is very important. So we have added a discussion on the Jordan Green guide as well as a comparison of the socio-cultural indicators found in this and other green building guidelines. (Lines 46-60 and Table 1)
Point 2: In your discussion on vernacular architecture (lines 70-74), a few examples of what this is might be helpful to the audience..
Response 2: Thank you, we have added an example of how vernacular architecture can provide such lessons. (Lines 93-101
Point 3: In the paragraph about decentralising decision making (lines 75-80), you might want to consult some of the literature regarding charrettes.
Response 3: Thank you for your comment we have tried to add a paragraph regarding this with suitable references. (Lines 106-114)
Point 4: Table 2 has some ambiguity within its design. It is not clear where the aesthetic and visual impact section ends and the role of privacy and dwelling technology begins.
Response 4: We appreciate the apparent ambiguity in this table. The indicators were presented as identified by the participants in the primary stages. They were consulted on the themes and their views against each theme were as shown. We thought to maintain the integrity of the findings and the reason was justified in a separate paper and is outside the scope of this one. Table 3 presents the resolved findings. Therefore, this table has been removed.
Point 5: Some of the typologies within this table are a little confusing as well. For example, why are the location of parking spaces part of the social aspects?
Response 5: Shared pathways and shared parking spaces in a community were discussed by participants in that study as a means to increase the potential for social interaction between residents. Tables 2 and 3 are cited from a previous paper and are better explained and discussed there. We have removed them from this paper in order to keep it concise and eliminate any duplicity and confusion.
Point 6: Figure 1 looks to be just a visual re-orientation of the material presented in Table 2...can you confirm that? It actually looks fine, but I am just wondering if it is a redudancy.
Response 6: You are right. It is a visual representation for relationship deduced from the first phase of the study and Table 2. Figure 1 outlines the research design for this paper. By bringing attention to the unspoken indicators of physical space, embodied sustainable elements of the home and what people said about their homes, all of which are central to an eco-cultural approach. We have added a small paragraph to explain this. (Lines 169-178)
Point 7: Also, Table 3 seems to be an expansion of Table 2; however, you seem to have left off the lower half of the categories mentioned within the earlier table.
Response 7: Thank you. The aim at the time was to establish how socio-cultural and environmental indicators correlate. The lower half of the indicators from Table 2 are displayed in the ecological advantage column in Table 3. However, following reviewers’ comments and suggestions, this table has been removed to eliminate any confusion as it can be found in our previous paper alongside better discussion.
Point 8: The presence of the term "modified indicators" in Table 4 requires further explanation. Was this concept mentioned in Jordan's Green Guide, or not? If so, how was it modified?
Response 8: Thank you. Further explanation was added to clarify this point. (Line 187-190).
Point 9: From lines 174-177, the awarding of points seems somewhat arbitrary. Please explain the rationale for the scoring hierarchy here.
Response 9: Thank you for your comments. We agree that the text didn't justify the scoring adequately. Justification has been added, we hope that this is now satisfactory. (Lines 208-214).
Point 10: Figures 2 and 10 seem to have some issues with visual quality.
Response 10: Better resolution figures were added.
Point 11: When considering the criteria for the three main dimensions of the tool evaluation, it would appear that "c" would be highly correlated with "a" and "b".
Response 11: Thank you for this observation. Yes this is the case. However, we also thought to distinctly and separately examine the extent to which this tool (and its inherent approach) is acceptable, compared to the existing approach represented in the Jordanian Green Guide. We have now highlighted results and findings to align with the three evaluation criteria in the conclusion (Lines 424-490 and lines 252-256).
Point 12: In Table 6, it would appear that item 1 has a statement that requires reverse coding compared with the other statements from the survey, based on the Likert scaling explained in the section above it...is this correct?
Response 12: Thank you for your question. Although some studies include reverse or negative coded words in their scales, we did not feel there is a need to do so. Negative worded or coded statements are intended to encourage respondents to read all items carefully rather than use a set pattern of responding (See Scale Development Theory and Applications Robert DeVellis). Our evaluation sessions were conducted face to face, and there were no concerns of participants answering positively worded statements as negative. Results from the "Users need additional support to use this tool for the first time" correlate with results from other effectiveness statements such as "the tool is easy to use" and "the tool design in pleasant" which showed that participants of the study faced an issue with tool usability aspects. (Lines 333-337 and lines 412 -495).
Point 13: While the manuscript overall is pretty well-written, there are still some issues related to spelling that can be properly addressed by going through the document a little more thoroughly.
- The word "average" in Tables 4 and 5 is just one example of this. On line 158, replace with "Table 4 presents..."
- On line 216, please add the word "and" before "how much time"
- On line 225, please address the phrase "draft of the final draft".
- On line 229, I believe that you meant to use the word "experience".
- You have the term "discussion" in both sections 4 and 5.
- On line 389, the phrase "else of its origin country" seems awkward.
Response 13: Thank you. These errors were addressed and further proofreading was made to the manuscript